# Academic Self-Concept Dramatically Declines in Secondary School: Personal and Contextual Determinants

**DOI:** 10.3390/ijerph19053010

**Published:** 2022-03-04

**Authors:** Álvaro Postigo, Rubén Fernández-Alonso, Eduardo Fonseca-Pedrero, Covadonga González-Nuevo, José Muñiz

**Affiliations:** 1Department of Psychology, University of Oviedo, Plaza de Feijoo s/n., 33003 Oviedo, Spain; postigoalvaro@uniovi.es (Á.P.); gonzalezvcovadonga@uniovi.es (C.G.-N.); 2Department of Educational Sciences, University of Oviedo, Aniceto Sela s/n., 33007 Oviedo, Spain; 3Department of Education of the Principality of Asturias Government, Plaza de España 5, 33007 Oviedo, Spain; 4Department of Educational Sciences, University of La Rioja, Luis de Ulloa 2, 26004 Logroño, Spain; eduardo.fonseca@unirioja.es; 5Nebrija University, Rector, Santa Cruz de Marcenado 27, 28015 Madrid, Spain; jmuniz@nebrija.es

**Keywords:** academic self-concept, grade retention, gender, immigrant condition, socio-economic index

## Abstract

Academic self-concept is one of the most important non-cognitive variables in determining students’ attitudes towards school and their performance. The objective of this study was to use a longitudinal approach to analyze how academic self-concept changed between primary and secondary schools and to analyze the factors that affected that progression. The sample consisted of 7379 students (47.4% girls) evaluated at two time-points: fourth grade and eighth grade. Six schooling pathways were analyzed: repeating a year before fourth grade, repeating between fourth and eighth grade, and repeating eighth grade. Five two-level hierarchical linear models of intrasubject means were assessed. The results indicate that academic self-concept falls dramatically between primary school and secondary school, varying according to background variables. Nevertheless, the most influential factor was the students’ schooling pathway. This study reinforces the evidence that, at least in the Spanish context, educational policies need to address alternatives to repetition.

## 1. Introduction

There is solid evidence indicating that academic self-concept—students’ perceptions of their academic abilities—is associated with variables such as educational results, intelligence, neuropsychological maturity, motivation, creativity, and empathy (e.g., [1,2,3,4,5,6,7,8,9,10,11,12,13,14]). Three models have been developed based on how the relationship between academic self-concept and school performance is understood. The first proposes that academic self-concept affects school performance. The second is the opposite and proposes that school performance affects academic self-concept. The third is the bidirectional model in which they mutually reinforce each other, and it is this final model that has the most support from research [2,6,7,10,12]. As in other psychological constructs, the formation and development of academic self-concept are affected by both personal and contextual variables [15]. In the case of academic self-concept, socio-cultural factors become significant given that it is constructed through a process of social comparison of one’s academic competence with others’ or with the class-group [16,17,18,19,20]. Although social comparison seems to be the most plausible explanation, studies have also shown that various background variables influence beliefs about one’s academic competence. These include gender [21,22,23,24,25,26], socio-economic level [27,28,29,30,31], and whether one is an immigrant [32,33,34,35,36,37], which should be considered in studies about academic self-concept.

Studies indicate a general decline in academic self-concept throughout students’ schooling [23,38,39], which has been confirmed by a meta-analysis of longitudinal studies [40]. For example, [41] found a significant fall in motivational variables as students progressed throughout their schooling, reporting that math and language academic self-concepts were among the noncognitive variables with the most significant declines. However, they were unable to confirm a fall in the general academic self-concept. Similarly, longitudinal studies have shown differential effects in the progression of self-concept depending on background factors: gender [21,23,42,43,44], socio-economic and cultural levels [45,46,47], and being an immigrant [48,49,50]. 

Another variable that may significantly influence academic self-concept is school repetition. The practice of repeating a school year varies widely between countries. While repeating a year in some countries is exceptional and only affects a tiny minority of students, it is widespread in others (e.g., [51,52,53,54,55]). The authors of Ref. [56] concluded that these differences in how widespread this measure was reflected well established the shared social beliefs about the effects of repetition on academic achievement and psychosocial outcomes. In countries with higher repetition rates, teachers, families, and educational authorities share the idea that repeating a year is beneficial for a student’s educational process. Nonetheless, the effects of repetition are debatable [57,58,59], even more so when looking at the benefits of repetition on socio-emotional outcomes in general and academic self-concept in particular. Research has produced mixed and even contradictory results [59,60,61,62,63,64]. One strand of research defends repetition as an incentive to improve as a student, as it would increase self-confidence, motivation, a love of learning, perseverance, and academic resilience, improving wellbeing at school and encouraging new personal relationships [58,65]. More than a few studies have found repetition to positively affect motivational factors, reporting improvements in repeaters’ engagement and self-concept in the short- and medium-term periods [18,60,63,66,67]. These studies are consistent with theories explaining the development of academic self-concept (see, [20]), which include the big-fish-little-pond effect (BFLPE) [17,68]. The generic explanation is that the student repeating a year is surrounded by fewer expert classmates (a year younger) who have lower levels of knowledge and skills, which allows the repeater to make a more positive evaluation of their own skills in comparison, thus improving their academic self-concept.

On the other hand, it is also reasonable to think that repetition has emotional and motivational costs, such as a worsened attitude towards school, loss of confidence in oneself, stress, and frustration [61,62,69,70,71,72], which increase the likelihood of dropping out of school [73,74,75]. In addition, those who repeat school years can be stigmatized by their new teachers and classmates, aggravating socio-affective costs [67]. This argument is not without support, as [62] studied the effect of repetition on the progression of motivational results (self-concept and academic interests and motivation to learn) in German sixth-grade students. The results showed a notable reduction in all motivational variables during the final few months in the original class, right before repetition. This drop was maintained the following year, but recovery occurred two years later.

In this context, the present longitudinal study has three main objectives: first, to determine how levels of academic self-concept change over time between primary school (fourth grade) and secondary school (eighth grade); second, to evaluate the role of sociodemographic variables (gender, nationality, and socio-economic level) and schooling pathways on the makeup of academic self-concept in primary education; and, finally, to analyze how these variables influence the progression of academic self-concept throughout schooling. A longitudinal design approach using a large sample will help shed light on this issue.

## 2. Materials and Methods

### 2.1. Participants

A total of 7379 students participated, which was the 4th grade student population in 2009 in the Principality of Asturias (Spain). The students were assessed at two time points: T_1_, when they were in 4th grade, and T_2_, when they were in 8th grade. The mean age at T_1_ was 9.6 years old (SD = 0.42); at T_2_, it was 13.78 (SD = 0.82). Just under half (47.4%) were girls, and 92.1% had Spanish nationality.

### 2.2. Instruments

#### 2.2.1. Academic Self-Concept

This was assessed using a questionnaire with five Likert-type items, and each had four response options: (1) I learn the lessons easily; (2) I get good grades; (3) I am a good student; (4) teachers consider me a good student, and (5) my family considers me a good student. The scale produces a score for *Academic self-concept* for each student from 0 to 3, with values close to 1 indicating low levels and scores around 2 indicating good academic self-concept. All items were directly worded to avoid psychometric biases [76,77]. At T_1_, the scale was essentially unidimensional: The first factor explained 66.55% of the variance, optimal implementation of parallel analysis [78,79] recommended a single dimension, and the indices of unidimensionality supported treating data as essentially unidimensional (UNICO = 0.999, ECV = 0.982, MIREAL = 0.090) [80]. In addition, there were excellent indices of fit (CFI = 0.999; RMSEA = 0.011; RMSR = 0.008). At T_2_, the data were similar: The first factor explained 72.84% of the variance, parallel analysis suggested a single dimension, and the indicators of unidimensionality (UNICO = 0.998, ECV = 0.959, and MIREAL = 0.155) and model fit (CFI = 0.996; RMSEA = 0.060; RMSR = 0.023) were good. Finally, despite the small number of items, there was high internal consistency at both T_1_ (*α* = 0.87; *ω* = 0.88) and T_2_ (*α* = 0.91; *ω* = 0.91). This scale has also shown adequate evidence of convergent validity with variables such as academic expectations and effort [14].

#### 2.2.2. Background Variables

Two dichotomous variables were considered: Gender (0 = male; 1 = female) and student *Nationality* (0 = Spanish; 1 = other). Socio-economic level was assessed using the *family’s Socioeconomic and cultural index* (ISEC) based on parents’ educational attainment and professions. To ensure data quality, each participating student’s form tutor reported information about parents’ education and professions. This information was used to create an approximately normal scale with a mean of 0 points and standard deviation 1 [*N*(0, 1)]. By using Exploratory Factor Analysis, ISEC was determined to be essentially unidimensional for the following statistical reasons: Optimal implementation of parallel analysis [79] recommended a single factor, the percentage of variance explained by the first factor was high (61.41%), indices of unidimensionality were suitable (UNICO = 0.952; MIREAL = 0.307) [80], and the indices of fit to the unidimensional model were very good (CFI = 0.977; RMSR = 0.049). The reliability in the current research was good (*α* = 0.79; *ω* = 0.80).

#### 2.2.3. Repetition and Schooling Pathway

Data from school administrations allowed us to create three dichotomous variables indicating when a student had to repeat a school year: *Rep_BeforeT_1_* (1 = repetition before 4th grade); *Rep_BetweenT_1_–T_2_* (1 = repetition during the four school years between T_1_ and T_2_); and *Rep_AfterT_2_* (1 = repetition at the end of 8th grade). These three variables were used to establish six school pathways (Figure 1) defined by the number of repetitions and when they occurred:0.Pathway 1: *Normal progress*. The student progresses through their schooling adequately in line with Spanish norms. They begin at age 9 in 4th grade primary education (T_1_ in the study) and start 8th grade (compulsory secondary education) at age 13 (T_2_). When they complete the school year corresponding to T_2_, they progress to the 9th grade. This profile includes the majority (71%) of students.1.Pathway 2: *First repetition on completion of 8th grade*. This profile has the same characteristics as pathway 1, except that the student repeats the year at the end of the 8th grade, starting the following year at the same level. For these students, the self-concept measure used at T_2_ is estimated during the year they are repeating. This group represented 3.9% of the students.2.Pathway 3: *Repetition between 4th and 7th grade*. These students were 9 years old at T_1_; however, they repeated a school year in one of the subsequent years, meaning that, at T_2_, they were 14. This group covered 16% of the students.3.Pathway 4: *Early repetition*. These students repeated a school year before they began 4th grade, and so at T_1_ they were 10 years old. They did not repeat any school years between the two time-points and, at T_2_, they were 14 years old. At the end of that school year, they progressed to 9th grade. This group represented 3% of the students.4.Pathway 5: *Early repetition and second repetition in 8th grade*. This group had the same characteristics as students in pathway 4, except that, at the end of the 8th grade they repeated the year, and so at T_2_ they were 15 years old. As in pathway 2, self-concept was assessed during the year they were repeating. This was the smallest group, with 1.1% of the students.5.Pathway 6: *Early repetition and second repetition between 4th and 8th grade*. These students had two repeated years, the first at some time before they began 4th grade, meaning that they were 10 years old at T_1_. Subsequently, they repeated another year, meaning that, at T_2_, they were 15 years old. At the end of 8^th^ grade, they progressed to 9th grade, as in Spain, students can only repeat two years during compulsory education. This group represented 5% of the sample.

### 2.3. Procedure

This study used data from the Diagnostic Educational Evaluation Program for the Principality of Asturias (Programa Evaluación de Diagnóstico Educativo del Principado de Asturias), which is an annual evaluation of all students in 4th and 8th grade compulsory education. All schools in the region are required by law to take part. Schools inform students’ families of the planned dates, and the families can refuse to participate, although that is relatively rare, meaning that it is a universal evaluation in practice. The test instruments are adapted for students with physical or motor deficiencies. Each school director is responsible for organizing the tests following the instructions laid down by the regional Department of Education. Quality control is the responsibility of the school’s inspection service, and the Department of Education performs coding and data analysis. The school principals manage and coordinate paper-based applications within their schools, and the school inspectorate monitors quality. The student context questionnaire, which included items making up the Academic self-concept, was applied in the same conditions at the two time points (4th and 8th-grade compulsory education).

### 2.4. Data Analysis

First, we calculated descriptive statistics and Pearson’s correlations for each of the variables used in the study. Subsequently, in pursuit of the study objectives, we specified a sequence of five hierarchical-linear models of intrasubject measures in two levels. Level 1 included the two measures of student self-concept, and level 2 included the sociodemographic variables and the repetition variable. For the first objective, we specified a fixed intercept and slope model (model 1), which only included the time-point variable (T_1_ vs T_2_), allowing the difference in mean academic self-concept at the two time-points to be estimated for the sample as a whole. We specified two random intercept, fixed-slope models for the second objective (variables affecting academic self-concept in primary education). Model 2 included sociodemographic variables, allowing the comparison of differences in academic self-concept at T_1_ in terms of those variables. Model 3 added an indicator of early repetition and provided an estimation of the level of students’ academic self-concept at T_1_ for students who had repeated a year before the 4th grade. The third objective was addressed with the final two models specified with random intercepts and slopes. Model 4 included sociodemographic variables, providing a picture of how academic self-concept changed between T_1_ and T_2_ in relation to those variables. Finally, model 5 included the slopes of the three repetition variables, allowing us to estimate the change in academic self-concept according to the number of repetitions and when they took place. Analysis was performed using HLM 7.0 [81].

The amount of missing data was small, ranging from 0 to 12%. We followed a two-stage process to recover it. When a case had incomplete data in some variable, the missing data was replaced with the mean from the same subject. If the values of the variable were completely missing, we used the expectation-maximization algorithm with auxiliary variables offered by SPSS 24 [82]. The authors of Ref. [83] showed that this two-stage process is the best for recovering missing data when, as in this case, the amount of missing data is small and the bias of the missing data (Not missing at random) is moderate.

## 3. Results

Table 1 shows that the correlation between the two measures of academic self-concept was moderately high and that, in general, the correlation matrix produced a consistent picture. Most repetitions were between the two time points.

Table 2 shows the results of the hierarchical regression models. Model 1 shows that, at T_1,_ the student population exhibited satisfactory academic self-concept, γ00 = 2.24, on a scale from 0 to 3. Four years later, the prediction fell by almost 23% (γ10 = −0.51 points). The variance components of this model were as follows: total variance was 0.495 points, of which 0.280 points (56.6%) corresponded to variations in student self-concept between T_1_ and T_2_. The remaining variance (0.215 points, 43.4%) was level 2 variance, corresponding to differences in the level of self-concept between students.

The next two models confirmed that both the sociodemographic variables and early repetition had significant effects at T_1_ (the second study objective). Model 2 shows that girls had greater academic self-concept and that immigrant students and students with low ISEC exhibited lower self-concept in 4th grade. The model explained almost 16% of the differences in student academic self-concept. Model 3, which addressed the effect of early repetition on academic self-concept at T_1_, predicted that students who had repeated a school year before T_1_ would have an academic self-concept that was 20% lower (−0.42 points) than non-repeating students once sociodemographic background variables were controlled for.

The final two models examined the influence of sociodemographic variables and the schooling pathways in the progression of academic self-concept (third study objective). Model 4 shows that the general decline in academic self-concept (γ10 = −0.52) was shallower in students with high ISEC (γ11 = 0.07), whereas the gender slope was not statistically significant. The slope was positive for immigrant students, although only marginally significant (γ13 = 0.06, *p* < 0.1), indicating that these students had similar falls in academic self-concept to Spanish students. The fifth model shows that, once sociodemographic variables were controlled for, the fall in academic self-concept seems to be linked to the point at which repetition happens. Students who were repeating the year at T_2_ exhibited a steeper negative slope than those who repeated a year between two time-points or those who repeated a year prior to T_1_, for which its slope was not statistically significant. It is also interesting that immigrant students demonstrated a positive slope between T_1_ and T_2_ (γ13 = 0.20).

Figure 2 shows the expected progression of perceptions of students’ abilities in six schooling pathways once background variables are controlled. Only the students who had not repeated any years (Pathway 1) maintained levels of academic self-concept that were close to satisfactory at T_2_. Repeating a year between T_1_ and T_2_ (Pathway 3) or repeating the year at T_2_ (Pathway 2) predicted falls of 0.44 and 0.63 points, respectively. In contrast, the slope for those who had early repetition but did not repeat any subsequent years (Pathway 4) was the same as Pathway 1, and the difference in self-concept between these students and non-repeating students was the same at T_1_ and T_2_ but did not worsen. In fact, these students had the best academic self-concept of those who had to repeat a year. Finally, students who repeated two years (Pathways 5 and 6) demonstrated unsatisfactory levels of academic self-concept, which is less than encouraging in terms of completing compulsory education.

## 4. Discussion

Using a longitudinal design, we assessed the progression in the level of students’ academic self-concept over a four-year period from primary (fourth grade) to secondary (eighth grade) school, examining the personal, school, and contextual variables that affected this change. In terms of the first objective of the study, the results show (model 1) a clear fall in academic self-concept between primary (fourth grade) and secondary (eighth grade). This is consistent with previous empirical research [23,38,39] and review studies [40,41] that indicate a decline in motivational variables in general and academic self-concept in particular as schooling progresses.

The second objective was to assess the effects of sociodemographic variables and early repetition of school years on academic self-concept in primary school. Model 2 showed that the background variables affect self-concept and have probably induced effects from the beginning of schooling or at least from early ages, which is in line with the results from [3,84,85]. In fourth grade, girls exhibited higher levels of self-concept than boys, which agrees with previous studies in similar age groups using general measures of academic self-concept [22,24]. This model also shows that immigrant students had clearly lower levels of academic self-concept. Bearing in mind the standard deviation of the criterion variable at T_1_ (0.64 points), the difference expressed as Cohen’s d would be 0.45 points (moderate effect), which is in agreement with previous studies [34,35]. Similarly, ISEC indicates that students from families with higher socio-economic and socio-cultural levels demonstrated higher levels of self-concept in fourth grade, again in line with previous studies [1,14,28,29,31,86]. Model 3 shows that academic self-concept was significantly lower in students who repeated a year prior to fourth grade, as other studies have also found (e.g., [69,71,72]). In fact, in our model, the effect size of early repetition on self-concept, was five times that of gender and twice that of nationality.

The final two models examined the effects of sociodemographic variables and schooling history on the progression of self-concept (third objective). Model 4 indicates that the regression slope of the gender variable was not significant in the four years after T_1_, which suggests that the fall in self-concept was similar in both genders and, therefore, that girls maintain their initial advantage over boys, which does not grow or shrink [21,44]. We also saw that students from high socio-economic and socio-cultural backgrounds exhibited shallower declines in self-concept between fourth grade and eighth grade, which is in line with previous research [45,46]. For immigrant students, the random slope models show that their initial disadvantage, at fourth grade, was tempered to the point where, in model 5, the slope becomes positive and significant, indicating that, at T_2_, immigrant levels of academic self-concept matched that of Spanish students. This has been documented in previous studies [36,37] and confirms that the schooling pathways and opportunities for progression and remaining within the educational system can compensate for initial inequalities.

Finally, model 5 allowed us to examine the influence of the schooling pathways on the progression of academic self-concept once their sociodemographic background was controlled for. We defined six pathways through schooling based on how many times and when students had to repeat a school year. The overall result, as expected based on previous research on the subject (e.g., [61,62]), was that repeating a school year negatively affected academic self-concept. Repetition was associated with a fall in self-concept from the moment it happened, and the more recent it was, the greater the fall [62]. Of the six pathways, the two that were associated with the worst progression of academic self-concept were Pathways 5 and 6, where students had repeated two years. The students who were repeating the year at the time of testing (T_2_) had the greatest fall. In contrast, students on Pathway 4, early repetition, exhibited a lower academic self-concept at T_1_ (in comparison with their non-repeating classmates), but that difference remained unchanged over the subsequent four years. These results are in line with previous studies indicating that repeating school years results in worse academic self-concept [61,62,72] and, hence, do not confirm research that indicated the opposite [18,58,63,66] or the big-fish-little-pond effect theory [68], which claimed that repeating students would improve their academic self-concept by comparing themselves to others with worse performance.

This study reinforces the evidence that, at least in the Spanish context, educational policies need to address alternatives to repetition. Firstly, clear standards that effectively define success expectations and offer the advantage of external control for teaching and learning processes are required [87]. Although international studies indicate that educational systems with high levels of repetition tend to demonstrate poorer performance [52], more nuanced analysis [88] has shown that OECD countries with very similar results in PISA exhibit very different rates of repetition. In other words, it seems that the criteria for assessing school progress do not have an objective foundation that is internationally comparable but instead rely on socially shared ideas and beliefs about the supposed benefits of repetition [56].

Secondly, research has supported the idea that early intervention helps students to successfully prepare for school and acquire strategies for success [87,89]. Hence, personal tutors and mentoring may be of great use, where the student has someone who can help them academically and especially emotionally [90,91]. In this regard, more emotionally related interventions teaching the student a better growth mindset about their own skills may benefit their emotional results [92,93,94]. In addition, because many problems begin in kindergarten [95], more preventative approaches should be considered, such as the pre-kindergarten program, which offers a child-centered curriculum with strong literacy and language development while working hand-in-hand with parents [96].

Implementing programs in schools to extend learning time can successfully affect performance, be it after school programs or even summer courses [97,98,99,100]. In addition, grouping different ages of students together in the classroom (an age range of more than 1 year) allows students to continually progress and learn at their own pace [87], calling for differentiated teaching as a main topic during teacher training [62,101]. Finally, continued education by teachers (where the same teacher teaches the same students over various school years) and continued teaching (where the students attend school for the same number of days, but with shorter breaks) have shown promising results for students at risk of repeating school years [5,87].

This study must be considered in light of its limitations. It uses a correlational model, which means that causal relationships cannot be established. Academic self-concept was assessed using a self-report, meaning that future studies should use other complementary measures. In addition, while the sample was very large, it was from a population in one region in the north of Spain, which means caution must be used if generalizing the results, nationally or internationally. Finally, in the future, it would be useful to use alternative methods such as propensity matching scores to perform complementary exploration of the data.

## 5. Conclusions

This study assessed students’ academic self-concept progression over four years from primary (fourth grade) to secondary (eighth grade) school, examining the personal, school, and contextual variables that affected this change. First, the results show a clear fall in academic self-concept between primary (4th grade) and secondary (8th grade). Second, background variables (sex, immigrant condition, and ISEC) affect self-concept and have probably induced effects from the beginning of schooling. Third, repeating a school year had some negative effects on academic self-concept. Thus, given the high number of retained students in Spain, exploring why students repeat (beyond medical, cognitive, or motor problems) is necessary. In summary, this study indicates that repeating school years results in worse academic self-concept; hence, repetition does not confirm the big-fish-little-pond effect theory, which claimed that repeating students can improve their academic self-concept by comparing themselves to others with worse performance. Therefore, our results question the efficacy of grade retention, at least from the point of view of the academic self-concept.

## Figures and Tables

**Figure 1 ijerph-19-03010-f001:**
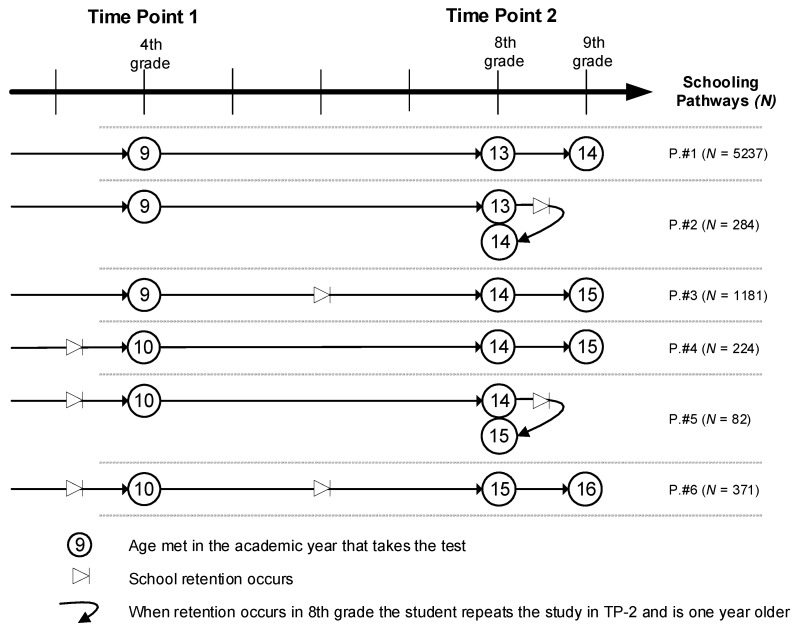
Schooling pathways.

**Figure 2 ijerph-19-03010-f002:**
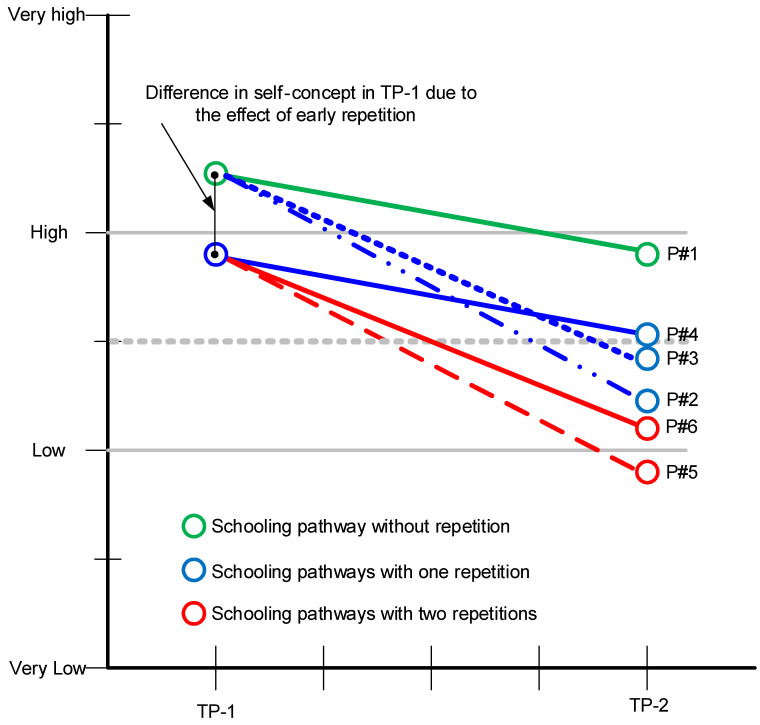
Progression of academic self-concept in relation to schooling pathways once background factors are controlled for.

**Table 1 ijerph-19-03010-t001:** Descriptive statistics and Pearson’s correlations.

	M	SD	1	2	3	4	5	6	7
1. Self-concept T_1_	2.24	0.64	-						
2. Self-concept T_2_	1.74	0.75	0.447	-					
3. Socioeconomic and cultural index (ISEC)	−0.01	0.94	0.227	0.289	-				
4. Gender (1 = girl)	0.47	0.50	0.065	0.060	−0.001	-			
5. Nationality (1 = Non-Spanish)	0.08	0.27	−0.164	−0.124	−0.144	0.008	-		
6. Repetition before T_1_ (1 = yes)	0.09	0.29	−0.217	−0.230	−0.250	−0.028	0.217	-	
7. Repetition between T_1_–T_2_ (1 = yes)	0.22	0.41	−0.363	−0.347	−0.284	−0.068	0.201	0.239	-
8. Repetition at the end of T_2_ (1 = yes)	0.05	0.22	−0.098	−0.208	−0.126	−0.040	0.057	0.109	−0.110

**Table 2 ijerph-19-03010-t002:** Adjusted hierarchical regression models for academic self-concept between time-point 1 and time-point 2.

	Model 1	Model 2	Model 3	Model 4	Model 5
Self-Concept (T_1_, γ00)	2.24 (0.01) *	2.22 (0.01)	2.26 (0.01)	2.26 (0.01)	2.26 (0.01)
*Intercepts (T_1_)*					
ISEC (γ01)		0.16 (0.01)	0.13 (0.01)	0.10 (0.01)	0.10 (0.01)
Girl (γ02)		0.09 (0.01)	0.08 (0.01)	0.08 (0.01)	0.08 (0.01)
Non-Spanish (γ03)		−0.29 (0.02)	−0.21 (0.03)	−0.24 (0.03)	−0.25 (0.03)
RepBeforeT_1_ (γ04)			−0.42 (0.02)	−0.42 (0.02)	−0.36 (0.02)
*Slopes*					
T_1_–T_2_ (γ10)	−0.51 (0.01)	−0.51 (0.01)	−0.51 (0.01)	−0.52 (0.01)	−0.40 (0.01)
ISEC (γ11)				0.07 (0.01)	0.03 (0.01)
Girl (γ12)				0.01 (0.02) *^ns^*	−0.02 (0.02) *^ns^*
Non-Spanish (γ13)				0.06 (0.03) *^p^*	0.20 (0.03)
RepBeforeT_1_ (γ14)					0.04 (0.03) *^ns^*
RepBetweenT_1_–T_2_ (γ15)					−0.44 (0.02)
RepAfterT_2_ (γ16)					−0.63 (0.03)
*Percentage of variance explained*					
Inter-subject variance		15.81%	22.33%	21.86%	37.21%
Total variance		6.87%	9.70%	9.50%	16.16%
Deviance (Np)	29,969.3 (2)	29,237.9 (2)	28,924.7	28,881.3 (2)	28,163.3 (2)

Np = number of parameters. *ns* = not statistically significant. * = standard error of estimation in brackets. *p* = 0.075.

## Data Availability

The file with the study data is kept by the Department of Education of the Principality of Asturias Government (Spain).

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
