# Peer review of "Academic Self-Concept Dramatically Declines in Secondary School: Personal and Contextual Determinants"

_ijerph, 2022, doi:10.3390/ijerph19053010_

Round 1
Reviewer 1 Report
First of all, I enjoyed reading the paper and this is on an important topic- but the authors could really delve into this in much more depth---because there are myriad reasons for the decrease in academic self-concept over time- first, obviously the material and school, gets harder- the topics are more complex and intricate- and students are not always well prepared. Secondly, curriculum is not always carefully and closely followed from 4th to 8th grade- often it is haphazard and disrupted by things like COVID and other extraneous variables. Now, the biggest issue is the reason why students are retained---this needs to be examined and explored---and there may be good reasons why students are retained---hospitalization , surgery, medical- and other reasons such as head injury, brain damage, and also mental retardation or intellectual deficiency or a learning disability or vision or hearing impairment. When students are retained--there need to be good reasons articulated- and a diagnosis given- and explained to both students and parents- to simply say " You are going to be kept back a grade " is not adequate- when the student may have some type of specific problem. Testing is imperative and in this year 2022- we now have excellent intelligence and cognitive tests to alert us to low I.Q. and low intellectual functioning, and we have good valid, reliable visual motor and perceptual tests to alert us to visual processing concerns.
I think a few sentences in the conclusions section could rectify or fix these concerns.
Author Response
Thank you very much for your kind comments which contribute to improving the quality of the manuscript. In Spain, frequently, students are retained for other reasons rather than cognitive, medical, or circumstantial problems (such as the COVID-19 pandemic). However, we believe that you are right about the variability of why a student can fail and repeat a course, requiring a deeper exploration of the reasons. Therefore, following your suggestion, these concerns have been added in the conclusions section (p. 10).
Reviewer 2 Report
After a detailed review of the article, I consider that it has sufficient criteria of scientific quality. Nevertheless, and within the methodological framework, information about the design and validation process of the data collection instrument can be included. In the results section, the analysis performed and the data obtained could be explained in greater detail, since the description of the results is very scarce in most cases.
Author Response
Thank you very much for your helpful comments. Following your suggestion, some statements about the design and validation of the self-concept instrument (p. 3) and data collection instrument (p. 5) have been added. In addition, the English have been reviewed again by an expert.
Reviewer 3 Report
Paper deals with an actual topic: academic self-concept in relation to personal and contextual determinants. It is written in 13 pages, with 9 pages addressed to the research issue and 4 pages addressed to the used references (101 sources). The paper is written in an interesting, informative, very clear and systematic way.
Introduction gives relevant data on research associated with academic self-concept, but it lacks a theoretical framework, even though authors used a very large number of references. I would recommend to the authors to be more elaborative in the introduction on theory, definition and measurement of academic self-concept in order to let the reader deeply understand the topic.
Three main objectives are stated in proper manner. Materials and methods include enough data and procedure descriptions that are sound and precise. Results are shown in a transparent and appropriate way. Discussion is elaborative and reasonable. Conclusions state the core of the research problem and results. Although the conclusion is precise and sound, it is very short. I would recommend the authors to do a more elaborative conclusion, stating an overall overview of the paper (key theoretical and empirical issues) and to explain the contribution of the paper to a better understanding of the research area.
It was great to get to know the results of longitudinal research of this kind on this topic. I consider them very useful for pedagogic theory and practice.
Author Response
Thank you very much for your insightful comments and excellent suggestions. Following your suggestions, a deeper description and theoretical elaboration on the academic self-concept has been added in the introduction (p. 1). Also, based on your comments, a more elaborate conclusion has been developed based on the present study's findings (p. 10).